# COVID-19 Variants in Critically Ill Patients: A Comparison of the Delta and Omicron Variant Profiles

**Alberto Corriero** [1,*](ID)**, Mario Ribezzi** [2]**, Federica Mele** [3](ID)**, Carmelinda Angrisani** [3]**, Fabio Romaniello** [4]**, Antonio Daleno** [5]**, Daniela Loconsole** [6](ID)**, Francesca Centrone** [6](ID)**, Maria Chironna** [6](ID) **and Nicola Brienza** [2,*]

1   Unit of Anesthesia and Resuscitation, University of Bari Aldo Moro, Piazza G. Cesare 11, 70124 Bari, Italy
2   Department of Interdisciplinary Medicine—Intensive Care Unit Section, University of Bari Aldo Moro, Piazza G. Cesare 11, 70124 Bari, Italy; mario.ribezzi@libero.it
3   Department of Interdisciplinary Medicine—Section of Legal Medicine, Policlinico di Bari Hospital, University of Bari, 70124 Bari, Italy; fedemele1987@gmail.com (F.M.); carmelindangrisani@hotmail.it (C.A.)
4   Department of Biomedical Science and Human Oncology, University of Bari Aldo Moro, Piazza G. Cesare 11, 70124 Bari, Italy; fabio.romaniello24@gmail.com
5   Hospital Direction, Azienda Universitaria Ospedaliera Consorziale Policlinico Bari, Piazza G. Cesare 11, 70124 Bari, Italy; antonio.daleno@policlinico.ba.it
6   Department of Interdisciplinary Medicine—Hygiene Section, University of Bari Aldo Moro, Piazza G. Cesare 11, 70124 Bari, Italy; daniela.loconsole@uniba.it (D.L.); francesca.centrone@uniba.it (F.C.); maria.chironna@uniba.it (M.C.)
*   Correspondence: alberto.corriero@gmail.com (A.C.); nicola.brienza@uniba.it (N.B.)

**Abstract:** Background: Coronavirus disease is a pandemic that has disrupted many human lives, threatening people's physical and mental health. Each pandemic wave struck in different ways, infectiveness-wise and mortality-wise. This investigation focuses on critically ill patients affected by the last two variants, Delta and Omicron, and aims to analyse if any difference exists between the two groups. Methods: intensive care unit (ICU) COVID-19 consecutive admissions between 1 October 2021 and 31 March 2022 were recorded daily, and data concerning the patients' demographics, variants, main comorbidities, ICU parameters on admission, and the outcome were analysed by a univariate procedure and by a multivariate analysis. Results: 65 patients were enrolled, 31 (47.69%) belonging to the Omicron versus 34 (52.31%) to the Delta group. The mortality rate was 52.94% for the Omicron group versus 41.9% for the Delta group. A univariate analysis showed that the Omicron variant was associated with total comorbidities number, Charlson Comorbidity Index (CCI), pre-existing pulmonary disease, vaccination status, and acute kidney injury (AKI). In stepwise multivariate analysis, the total number of comorbidities was positively associated with the Omicron group, while pulmonary embolism was negatively correlated with the Omicron group. Conclusion: Omicron appears to have lost some of the hallmarks of the Delta variant, such as endothelialitis and more limited cellular tropism when it comes to the patients in the ICU. Further studies are encouraged to explore different therapeutic approaches to treat critical patients with COVID-19.

**Keywords:** SARS-CoV-2; SARS-CoV-1; COVID-19; COVID-19 variants; delta; omicron; disease severity; ICU; endothelial dysfunction





## 1. Introduction

The Coronavirus disease 2019 (COVID-19) is a global pandemic that has disrupted many human lives and dramatically changed the world, threatening different aspects of society. There have been 528,816,317 confirmed cases of COVID-19, including 6,142,735 deaths, reported to WHO as of 3 June 2022 [1].

Different waves of the pandemic by specific variants have characterised the last two years. Firstly, the Delta variant (B.1.617.2) was defined as a variant of concern (VOC) when the World Health Organization (WHO) stated in June 2021 that it was becoming the dominant strain globally. It was first detected in India in late 2020 and led to several

waves contributing to the infection spreading to over 179 countries by November 2021 [2]. It presented two mutations on the spike protein, both of which were present in the previous COVID-19 form [2]. The last wave in 2021, first affecting South Africa, was the one caused by the variant B.1.1.529 (Omicron) [3]. The first confirmed B.1.1.529 infection was discovered in a specimen collected on 9 November 2021 [3], and on 26 November 2021 the WHO declared Omicron a VOC [3]. Omicron has 37 mutations in the spike protein that affect how it spreads and the severity of the illness it causes. Compared with Delta, the Omicron variant appears to increase the risk of re-infection by 2.4 times. The high number of mutations can explain this finding [4]. Omicron is composed of several sublineages. BA.1, BA.1.1 (or Nextstrain clade 21K), and BA.2 (or Nextstrain clade 21L) are the most common [5]. In the report by "Istituto Superiore di Sanità" (ISS) on 25 March, in the previous 45 days up to 25 March, Omicron represented 98.3% of COVID-19 cases [6]. This 98.3% turned into 100% in the latest report on 31 May as it states that Omicron is virtually the only variant present in Italy. Omicron BA.2 is predominant compared with BA.1 as it represents 93.83% of the analysed samples [7]. No evidence shows a difference in disease severity between the two lineages [6].

Data comparing the severity of Omicron and Delta coming from different parts of the world, such as South Africa, Canada, Scotland, and England, suggest that the risk of hospitalisation or death is lower for Omicron cases compared with Delta cases [8–12]. This variant appears to be milder in all age groups, from adults older than 65 to children that lack vaccination [13]. This finding is supported by the fact that in hospitals, patients have been arriving less often with pneumonia-like symptoms but rather with clinical conditions that match the one of the common cold [14]. It is also worth mentioning that most of the Omicron cases reported were incidental, with people arriving at the hospital for other medical, surgical, or obstetric reasons and testing positive for the virus during routine screening on admission. Nevertheless, a relatively small percentage of patients required high care or intensive care unit (ICU) admission [11]. Currently, data on the spectrum of disease severity caused by the Omicron variant and prevalence of comorbidities compared with the Delta variant in patients admitted to the ICU are lacking. Only one recently published study serves this very purpose [15], as it shows that when patients are admitted to the ICU for pneumonia, the severity of the disease appears to be similar to that of Delta, with no difference in the risk of in-ICU mortality. A different virus variant can cause a more severe form of the disease in a patient with particular comorbidity that exposes that specific patient to a less favourable evolution of the pathology. The Omicron cohort can present in the critical care environment with a different subset of patients compared with what the Delta does, as a natural process of selection at work [16].

This investigation aims to assess differences in mortality, linked comorbidities, and general outcomes associated with Omicron versus Delta groups in patients admitted into COVID-19 ICU. This information can help to understand how this variant behaves and the new therapeutic approaches that can be adapted to fight the disease.

## 2. Materials and Methods

A prospective study was conducted on critically ill patients admitted because of COVID-19 between 1 October 2021 and 31 March 2022 in the COVID-19 ICU of Azienda Universitaria Ospedaliera Consorziale Policlinico Bari, Bari, Italy.

The parameters retrieved in all patients on ICU admission were age, sex, comorbid conditions, total number of comorbidities, Charlson Comorbidity Index (CCI), vaccination status, P/F ratio, acute kidney injury (AKI), catecholamine support, and pulmonary embolism. Moreover, ICU length of stay (ICU LOS), total hospitalisation (LOS), and outcome were assessed for each patient. CCI is a clinical tool to categorise patients' comorbidities based on the International Classification of Diseases (ICD). Each comorbidity category has a weight (from 1 to 6) assigned based on the adjusted risk of mortality or resource use. The sum of all weights results in a single comorbidity score for each patient, and a score of zero indicates no comorbidities discovered. The higher the score, the more likely the

predicted outcome is death or increased resource use [17]. Comorbidities include chronic illnesses that affected either the cardiovascular axis rather than renal, neurologic, metabolic, rheumatic, liver, or pulmonary axes. A history of previous or active malignancy was also noted. The cardiovascular axis includes diseases that affect the heart; primarily hypertension, previous myocardial ischemia, or cardiomyopathy. The pulmonary axis includes dysfunctions that primarily affected the lungs, such as chronic pulmonary obstructive disease (COPD), and pulmonary fibrosis. The metabolic axis includes comorbid conditions such as diabetes, obesity, metabolic syndrome, and other diseases disrupting the body's metabolism. The renal axis is focused on the presence of chronic renal failure. The liver axis includes diseases such as hepatic steatosis and other comorbidities that affected the liver chronically. The Neurologic axis includes peripheric neuropathies, progressive degenerative nervous system disorders such as Parkinson's disease and Alzheimer's disease, and any condition that affects the nervous system either centrally or peripherally.

The vaccination status includes patients who are either fully vaccinated with two doses or fully vaccinated plus a booster versus those who are not vaccinated or vaccinated with a single dose.

AKI was defined according to Kidney Disease Improving Global Outcome guidelines [18]. Catecholamine support was defined as continuous norepinephrine infusion of any dosage. Pulmonary embolism was interpreted as the presence of pulmonary embolism signs on a computer tomography pulmonary angiography performed on arrival in the ICU ward or prior to that.

For each patient, a nasopharyngeal swab (UTM, FLOQ Swabs TM, Copan Italia, Brescia, Italy) was collected and processed at the Laboratory of Molecular Epidemiology and Public Health of the Hygiene Unit (A.O.U.C. Policlinico Bari, Italy) which is the coordinator of the Regional Laboratory Network for SARS-CoV-2 diagnosis in the Apulia region. All samples were subjected to a commercial three-target multiplex real-time PCR assay based on the identification of the N, ORF1ab, and S genes (Thermo Fisher Scientific Waltham, MA, USA). One of the characteristic mutations of the Omicron VOC is the 69/70 deletion in the spike protein, which impairs the detection of the S gene (S gene target failure (SGTF)) [19]. This mutation is not present in the Delta variant, and therefore, SGTF was used as a proxy marker for identifying the Omicron VOC. Moreover, the SGTF-positive samples were confirmed as the Omicron VOC by using a commercial multiplex real-time PCR kit (Seegene Allplex SARS-CoV-2 variants I and II Assay, Arrows Diagnostics, Genova, Italy) as previously described [20].

*Statistical Analysis*

Descriptive statistical analysis was conducted with the program Medcalc for Windows v14.8.1. For continuous variables, data are expressed as mean ± S.D. or median and interquartile ranges (IQR), and as an absolute number or percentage, with an odds ratio (OR) and 95% confidence intervals (CI) for categorical variables. A univariate analysis was then used to compare the two primary groups (Delta versus Omicron) with all main variables. Two additional parallel univariate analyses were performed: vaccinated/unvaccinated Delta versus vaccinated/unvaccinated Omicron.

The unpaired Student *t*-test was used to compare continuous variables. To compare categorical variables, the Chi-square or Fisher exact test was used. A multivariate analysis was performed by a backward binomial logistic regression procedure [21], with patient variant as the dependent outcome variable. All variables, represented in all patients and with a *p*-value < 0.20 in the univariate analysis, were included in the model. The Hosmer–Lemeshow test assessed the power of the model goodness-of-fit test. A *p*-value of <0.05 was considered statistically significant.

## 3. Results

A total of 65 patients were enrolled in the study, 31 (47.69%) had the Delta variant versus 34 (52.31%) of them had the Omicron variant. The Omicron cohort's patients all

presented the Omicron BA.1 variant. Demographic data, vaccination status, and pre-existing comorbidities are presented in Table 1.

**Table 1.** Demographic data, vaccination status, and pre-existing comorbidities of patients belonging to Delta and Omicron groups on ICU COVID-19 admission. Values are means (±S.D.) or number (%). OR: odds ratio; CI: confidence interval.

| VARIABLES | DELTA | OMICRON | OR | CI 95% | *p*-Value |
|---|---|---|---|---|---|
| Number of patients (%) | 31 (47.69%) | 34 (52.31%) | | | |
| Age (years), mean (±SD) | 65.39 (11.36) | 65.79 (10.17) | | | 0.8793 |
| Sex (F/M) | 10/21 | 11/23 | 0.99 | 0.35–2.82 | 0.7969 |
| Vaccination, n (%) | 13 (41.94%) | 24 (70.59%) | 3.32 | 1.19–9.27 | 0.0376 |
| Total comorbidities, mean (±SD) | 2.26 (1.90) | 4.03 (2.83) | | | 0.0047 |
| CCI (points), mean (±SD) | 3.16 (1.97) | 4.20 (2.06) | | | 0.0409 |
| Cardiovascular axis, n (%) | 20 (64.5%) | 22 (64.7%) | 1.1 | 0.39–3.09 | 0.9344 |
| Pulmonary axis, n (%) | 3 (9.7%) | 12 (35.3%) | 5.09 | 1.28–20.29 | 0.0313 |
| Metabolic axis, n (%) | 11 (35.5%) | 17 (50%) | 1.82 | 0.67–4.92 | 0.3525 |
| Neoplastic disease, n (%) | 2 (6.4%) | 7 (20.59%) | 3.76 | 0.72–19.70 | 0.1975 |
| Liver axis, n (%) | 2 (6.4%) | 2 (5.9%) | 0.91 | 0.12–6.85 | 0.6735 |
| Rheumatic axis, n (%) | 3 (9.7%) | 4 (11.76%) | 1.24 | 0.25–6.06 | 0.897 |
| Renal axis, n (%) | 1 (3.2%) | 6 (17.65%) | 6.43 | 0.73–56.80 | 0.1408 |
| Neurologic axis, n (%) | 2 (6.4%) | 4 (11.76%) | 1.93 | 0.33–11.38 | 0.7564 |

In univariate analysis (Table 1), no statistical difference was found in age, sex, and single comorbidities except for pre-existing pulmonary disease, which had a significant difference (*p*-value = 0.0313). A statistically significant difference was found in vaccination status, the total number of comorbidities and CCI (Table 1). The number of vaccinated patients, the total number of comorbidities, and CCI were significantly higher in the Omicron group than in the Delta group.

ICU admissions data and outcomes are shown in Table 2.

**Table 2.** ICU clinical parameters on admission and outcome of patients belonging to Delta and Omicron groups. Values are means (+S.D.) or number (%). OR: odds ratio; CI: confidence interval.

| VARIABLES | DELTA | OMICRON | OR | CI 95% | *p*-Value |
|---|---|---|---|---|---|
| P/F ratio, mean (±SD) | 156.57 (65.98) | 157.31 (84.56) | | | 0.9709 |
| AKI, n (%) | 3 (9.7%) | 13 (38.24%) | 5.78 | 1.46–22.9 | 0.0172 |
| Catecholamine support, n (%) | 17 (54.8%) | 25 (73.53%) | 2.29 | 0.81–6.47 | 0.1887 |
| Pulmonary embolism, n (%) | 6 (19.3%) | 1 (2.9%) | 0.13 | 0.01–1.12 | 0.08 |
| ICU LOS (days), mean (±SD) | 17.84 (13.24) | 13.34 (12.67) | | | 0.1733 |
| LOS (days), mean (±SD) | 33.18 (18.61) | 22.59 (17.17) | | | 0.0652 |
| Deaths, n (%) | 13 (42%) | 18 (52.94%) | 0.64 | 0.24–1.71 | 0.523 |

No statistical significance was found in P/F ratio, catecholamine support, ICU LOS, Total LOS, and pulmonary embolism. A statistically significant difference (*p*-value = 0.0172) was found in AKI, being predominant in the Omicron group (38.24% vs. 9.7%). The mortality rate for the Omicron patients' group was 52.94%, while the mortality rate for the Delta patients' group was 41.9%. No statistically significant difference was found (*p*-value = 0.523). In the univariate comparison of vaccinated/unvaccinated Delta versus vaccinated/unvaccinated Omicron, only the variable neoplastic disease reached significative difference (*p*-value = 0.032) in the cohort Delta vaccinated versus Omicron vaccinated.

Among the ten variables, with a *p*-value < 0.20 in univariate analysis included in binomial logistic regression analysis, three reached statistical significance (Table 3). The total number of comorbidities and pulmonary embolism fell below the *p*-value threshold of 0.05. The goodness-of-fit test indicated that the sample data fitted well with a distribution from a population with a normal distribution (Hosmer–Lemeshow *p* = 0.75).

**Table 3.** Stepwise logistic multivariate analysis obtained the main differential factors among patients belonging to Delta and Omicron groups on ICU COVID-19 admission.

| Variables | OR | 95% CI | Binomial Logistic Analysis *p*-Value |
|---|---|---|---|
| Pulmonary embolism | 0.04 | 0.001–0.93 | 0.0453 |
| AKI | 3.32 | 0.76–14.50 | 0.1099 |
| Total comorbidities | 1.51 | 1.12–2.04 | 0.0068 |

## 4. Discussion

The primary findings of the current study show that Omicron patients admitted into ICU ward have a lower incidence of pulmonary embolism and a higher number of co-morbidities than the Delta group. The latter reflects as well in a raised CCI. Furthermore, pre-existing pulmonary comorbidity, positive vaccination status, and AKI were predomi-nant in the Omicron group. All Omicron patients presented the BA1 variant, as this variant was the main strain circulating in Italy during our investigation [6].

The incidence of ICU admission seems to decrease with the Omicron variant. This finding was recently confirmed by Vieillard-Baron et al., who discovered that Omicron patients are less likely to be admitted to the ICU compared with Delta ones [15]. Few studies have been completed in different countries where the Omicron severity and other attributes were compared with the Delta taking into account all hospital sections. As a general finding, a decreased severity was found in the Omicron group [8,10–13,22–25]. For example, Omicron infection had about a two-fold lower mortality relative to Delta using data from the USA's Delta and Omicron infection waves [24].

The reasons that explain this lower severity are not clear. However, research in a mouse model by the University of Liverpool [26] that is not yet peer-reviewed suggests that Omicron does not infect lung cells as efficiently as Delta variants and previous variants, which in turn makes it less damaging with a corresponding shift in milder symptoms. Another possible mechanism that can explain Omicron's reduced severity is that, in contrast to Delta, Omicron does not effectively inhibit the host cell interferon immune response, as Bojkova et al. found in their study [27]. The interferons are a group of proteins released by infected cells, which signal to the other system cells to resist viral growth; this is a critical mechanism in fighting the replication of many viruses, including SARS-CoV-2 [28]. It is also important to underline that measures of disease severity should be interpreted carefully because sometimes this can be misleading, as noted by Sigal et al. [24].

SARS-CoV-2 is four times more likely to cause severe illness in COPD or other respira-tory disorders; therefore, it is to be expected to find patients with these diseases more easily in a critical care setting. What can provide an edge to Omicron infected COPD patients compared with Delta infected COPD patients (as this is the case of this investigation where the pulmonary axis was more present in the Omicron group) is the fact that Omicron thrives more in the airways than in the lungs as shown by different studies [29,30]; hence, patients with COPD or other pulmonary diseases affecting the airways can have a higher risk of developing severe COVID-19.

In this study, the higher number of comorbidities and, therefore, higher CCI leads to a trend toward higher mortality (52.94% for Omicron versus 41.9% for Delta) even though it is not statistically significant, probably due to the restricted sample size; this pattern is, on the one hand, in line with data from other studies in the last two years that have shown that pre-existing comorbidities represent independent risk factors associated with COVID-19 in-hospital mortality [31–35]. On the other hand, no difference in the mortality between the two groups was found by Vieillard-Baron et al. However, only pneumonia and immunosuppression were considered to be comorbidities [15]. Therefore, Omicron might raise the mortality rate in ICU compared with Delta only when all comorbidities are considered. Further studies are advised to assess this possibility.

The next question is why patients with Omicron that reach the ICU have more comor-bidities than the Delta group. No other study supports this finding at the moment, as this

is among the first attempts to understand how differently Omicron affects ICU patients compared with Delta. It is hypothesised that vaccines offer better protection to Omicron patients than Delta ones. Only critical patients with a high number of pre-existing comorbidities require ICU admission. It has been seen that three vaccine doses raise protection in Omicron patients up to 60–75%, and considering the progression of the vaccine campaign during the Delta wave compared with the Omicron wave, the latter included more patients who had received at least one dose of vaccine or even a booster dose [36,37]. The more present positive vaccination status during the Omicron wave matches our findings, which can be linked to the vaccination campaign in Italy [38].

AKI is common in COVID-19 patients and is associated with a poor prognosis. Some studies have enlightened the role of acute tubular injury and direct renal tropism in the pathophysiology of COVID-19 AKI [39–42]. The latter can also explain why AKI is predominant in the Omicron group. Peacock et al. [43] suggested that one of the reasons why Omicron replicates more quickly than Delta and previous variants is that Omicron might have become less specialised in cell tropism by using the ubiquitous endosomal pathway for entry, allowing it to infect more types of cells. Therefore, it can be hypothesised that an increase in kidney tropism by Omicron can explain our finding.

The pulmonary embolism finding aligns with a recent retrospective French study on 13 academic hospitals in the Paris area from 29 November 2021 to 10 January 2022; 3.4% of the Delta cohort had signs of pulmonary embolism on Chest CT vs 1.2% of the Omicron cohort [44]. Furthermore, McMahan et al. demonstrated that in hamsters, the Omicron variant caused reduced endothelialitis compared with hamsters infected with previous variants [45]. These data might suggest a shift in the pathophysiology of COVID-19. In the past two years, different reviews and clinical findings have underlined the role of endothelial dysfunction in the contribution to severe COVID-19 [46–50]. Endothelial dysfunction pathogenesis can be classified as a direct viral effect, cytokine release, oxidative stress, coagulation disruption, and immune cell response. The cytokine release is referred to as cytokine storm and is the leading actor in the endothelial damage observed in severely ill COVID-19 patients [49].

This knowledge has led researchers and clinicians to explore many therapeutic drugs that might interfere with the endothelial side of the damage [47]. For this matter, as complement dysregulation is among the "main knights" of the cytokine storm, different complement inhibitors (anti C3 and anti-C5) have been used to soothe the storm and consequently diminish the endothelial damage; anticoagulants such as Low Molecular Weight Heparin and endothelial cell protectors such as Defibrotide have been used for the same purpose [49]. Anticoagulant therapy has been routinely recommended and shown to reduce mortality in COVID-19 patients, but this increases bleeding risk [51]. Hence, anticoagulant therapy is defined as a "double-edged" sword; on one side, it can reduce thrombotic complications due to endothelialitis. However, on the other side, it can lead to fatal complications such as spontaneous hematoma [52]. It might be supposed that the Omicron variant has modified pathophysiology compared with previous variants, sparing the endothelial side of the damage and the thromboembolism complication. In that case, this double-edged sword can turn into a single-edged blade, and the risk and benefit balance is shifted towards the patient's risk. It is also true that anticoagulants in the ICU are routinely used to prevent and treat clots. For prophylaxis usage, the dosage is lower than the current anticoagulant dosage used in most protocols for COVID-19 patients [53].

Our study has some limitations, including the small sample of patients due to the short time frame for collecting the data. However, this restricted temporal window can act positively by favouring the homogeneity of treatments and protocols adopted.

In summary, results indicate that in the critical care setting, the Omicron variant lost some hallmarks of the previous variants, such as endothelialitis, and gained or enhanced others, such as an extended cellular tropism. Furthermore, Omicron patients have more comorbidities than Delta ones, which can be linked to better vaccine protection in the Omicron cohort. These results should encourage further studies with larger sample sizes

to overcome the limitations of our study to explore new possibilities that might eventually lead to different therapeutic approaches to improve survival and reduce mortality of patients infected with Omicron. Different treatment strategies should be tailored to the patient's COVID-19 variant. It is evident that throughout the COVID-19 pandemic, the SARS-CoV-2 virus mutated [54], resulting in genetic variation in the population of circulating viral strains, which might also affect the clinical outcome of the disease. This matter becomes crucial as a new Omicron subvariant recently caused China to experiment with its most significant wave of COVID-19 infections since the pandemic's start [55].

**Author Contributions:** Conceptualisation, A.C., M.R. and N.B.; Methodology, A.C., A.D., D.L., F.C., M.C., M.R. and N.B.; Data collection, C.A., D.L., F.C., F.M., F.R. and M.C.; Statistical analysis, A.C, D.L. and M.C.; Writing original draft preparation, A.C., M.R. and N.B.; Writing review and editing, A.C., A.D., D.L., F.C., M.C, M.R. and N.B.; Supervision, M.C. and N.B. All authors have read and agreed to the published version of the manuscript.

**Funding:** This research received no external funding. Financing Not applicable, no funding was received to contribute to the preparation of this manuscript.

**Institutional Review Board Statement:** Not applicable.

**Informed Consent Statement:** The present study was based in the University of Bari (Italy), in full accordance with ethical principles, including the World Medical Association Declaration of Helsinki and the additional requirements of Italian law. Furthermore, the University of Bari, Italy, classified the study to be exempt from ethical review as it carries only negligible risk and involves the use of existing data that contains only non-identifiable data about human beings.

**Data Availability Statement:** All experimental data to support the findings of this study are available by contacting the corresponding authors upon request.

**Conflicts of Interest:** The authors declare no conflict of interest.

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
