# Peer review of "COVID-19 Variants in Critically Ill Patients: A Comparison of the Delta and Omicron Variant Profiles"

_2036-7449, doi:10.3390/idr14030052_

Round 1

Reviewer 1 Report

The authors compared the ICU parameters on admission and the outcome of enrolled 65 ICU COVID-19 patients by a univariate procedure and by a multivariate analysis. They found that :1) 31 patients (47.69%) belonging to the Omicron versus 34 patients (52.31%) to the Delta group. 2) The mortality rate was 52.94% for the Omicron group vs 41.9% for the Delta group. 3) The Omicron variant was associated with total comorbidities number, Charlson Comorbidity Index, pre-existing pulmonary disease, vaccination status and acute kidney injury (AKI). 4) The total number of comorbidities was positively associated with the omicron group, while pulmonary embolism was negatively correlated with the omicron group. Finally, they made a conclusion that Omicron appears to have lost some of the hallmarks of the Delta Variant, such as endothelialitis and more limited cellular tropism, when it comes to the patients in the ICU.

Major concerns:

1.        Were 65 enrolled patients vaccinated with the same kind of vaccine?

2.        If the authors can divide each variant virus infected patients into two groups, vaccinated group and unvaccinated group, and then compare the two groups (vaccinated/unvaccinated Delta versus vaccinated/unvaccinated Omicron) with all main variables, respectively, by a univariate procedure and by a multivariate analysis, the real Delta and Omicron variant profiles will be obtained.

Reviewer 2 Report

This clinical study compared the critically ill patients affected by the two SARS-CoV-2 variants Delta and Omicron, and analysed the difference  between the two groups. Although the sample of patients was small due to the short time frame for collecting the data, the authors made some valuable conclusions. The Omicron variant might have lost some hallmarks of the previous variants, such as endothelialitis and enhanced cellular tropism. The results encouraged further studies to explore the possibilities that might eventually lead to different therapeutic approaches to improve survival and reduce mortality of patients infected with Omicron. 

There is only one concern that the authors need to emphasize in the Discussion. The Omicron variant appears to be milder than Delta in all age groups as reported, which causes less ICU cases. In this study, the mortality rate for the Omicron patients' group was even a little bit higher than that for the Delta patients' group, which might be due to the fact that Omicron patients enrolled in this research were associated with more underlying diseases. The author should dicuss this point carefully to avoid any misleading.

Reviewer 3 Report

Overall, good work. There's a challenge though that the pandemic is moving faster than academic publishing. Correspondingly:

* Please update the Covid numbers of cases and deaths as of June 1 instead of April 1. 

* Please update with the latest numbers comparing BA.1 and BA.2 in terms of dominance and percentage of caseload. 

* Please do not capitalize the word "variant." 

* Importantly, please discuss how your study compares to the results found in the following studies: 

Lauring, Adam S., et al. "Clinical severity of, and effectiveness of mRNA vaccines against, covid-19 from omicron, delta, and alpha SARS-CoV-2 variants in the United States: prospective observational study." bmj 376 (2022).

Sigal, Alex, Ron Milo, and Waasila Jassat. "Estimating disease severity of Omicron and Delta SARS-CoV-2 infections." Nature Reviews Immunology (2022): 1-3.

Wrenn, Jesse O., et al. "COVID‐19 severity from Omicron and Delta SARS‐CoV‐2 variants." Influenza and Other Respiratory Viruses (2022).

Reviewer 4 Report

1. The English need improvement since there are some grammatical and syntax errors in the manuscript. For example,

·         in line number 19, the words “The Coronavirus” may be as “Coronavirus”;

·         in line number 26, “a multivariate” as “multivariate”;

·         in line number 28, “A univariate” as “Univariate”;

·         in line number 136, “odds” as “an odds”;

·         in line number 154, “total” as “the total”;

·         in line number 181, “Omicron” as “the Omicron”;

·         in line number 202, the words “therefore patients”.

The grammar mistakes which are not mentioned here are also to be checked and corrected properly.

3. Check the abbreviations throughout the manuscript and introduce the abbreviation when the full word appears the first time in the text and then use only the abbreviation (For example, acute kidney injury (AKI), Charlson Comorbidity Index (CCI), ICU etc.,). And it should be in both abstract as well as in the remaining part of the manuscript. Make a word abbreviated in the article that is repeated at least three times in the text, not all words need to be abbreviated.

3. The conclusion section appears to be just a detailed summary of results/observations. Moreover, the authors may be included the limitation of the present findings and future direction for a better understanding of the manuscript.

Round 2

Reviewer 1 Report

1.        This investigation aims of the authors are to assess differences in mortality, linked comorbidities, and general outcomes associated with Omicron versus Delta groups in patients admitted into COVID ICU, when the univariate analysis is applied to this study, the two groups to be compared can be comparable and reveal the hidden truth only if their basic conditions are consistent. In addition, ruling out other confounding factors is required, factors such as vaccination status and pre-existing comorbidities inevitably affect the severity extent and prognosis of COVID-19 patients. The cohort enrolled without vaccination and pre-existing comorbidities would be better to compare these 2 variants infection. 

2.        The authors show that no significant variable was found and it may be inconclusive when performing two parallel univariate analyses (vaccinated/unvaccinated Delta versus vaccinated/unvaccinated Omicron), in my view, it is valuable since it is the truth, even when there exists no significant difference between the compared paralleled groups. 

3.        line74-79, the reference 14 and 15, the conclusive results of their study should be elucidated clearly in this manuscript.

Reviewer 3 Report

Thank you for the much-improved updated manuscript!

Please note, though: You wrote that "Omicron BA.2 is predominant compared to BA.2" -- probably should be "Omicron BA.2 is predominant compared to BA.1" yes? 

Author Response

Dear reviewer.  

Yes the BA.2 was a typing mistake ,it has been fixed in the manuscript.

Thank you for your time

This manuscript is a resubmission of an earlier submission. The following is a list of the peer review reports and author responses from that submission.